# Trajectories of the Prevalence of Sarcopenia in the Pre- and Post-Stroke Periods: A Systematic Review

**DOI:** 10.3390/nu15010113

**Published:** 2022-12-26

**Authors:** Tatsuro Inoue, Junko Ueshima, Fumiya Kawase, Haruko Kobayashi, Ayano Nagano, Kenta Murotani, Yoko Saino, Keisuke Maeda

**Affiliations:** 1Department of Physical Therapy, Niigata University of Health and Welfare, Niigata 950-3198, Japan; 2Department of Nutrition Service, NTT Medical Center Tokyo, Tokyo 141-8625, Japan; 3Department of Nutrition, Asuke Hospital Aichi Prefectural Welfare Federation of Agricultural Cooperatives, Aichi 444-2351, Japan; 4General Incorporated Association Manabi Public Library, Aichi 465-0015, Japan; 5Department of Nursing, Nishinomiya Kyoritsu Neurosurgical Hospital, Hyogo 663-8211, Japan; 6Biostatistics Center, Kurume University, Fukuoka 830-0011, Japan; 7Department of Clinical Nutrition, Cancer Institute Hospital, Japanese Foundation for Cancer Research, Tokyo 135-8550, Japan; 8Department of Geriatric Medicine, Hospital, National Center for Geriatrics and Gerontology, Aichi 474-8511, Japan

**Keywords:** muscular atrophy, cerebrovascular disease, aged, nutrition therapy

## Abstract

Interventions for stroke-related sarcopenia in patients with stroke are needed, but the details of the target population are unclear. This systematic review aimed to identify trajectories of the prevalence of sarcopenia in the pre- and post-stroke periods and to determine the diagnostic criteria used in patients with stroke. We searched for literature in six databases: MEDLINE, EMBASE, Web of Science, Cochrane Central Register of Controlled Trials, CINAHL, and Ichushi-web (in Japanese). We included 1627 studies in the primary screening, and 35 studies were finally included. Of the 35 studies, 32 (91.4%) included Asian patients, and the criteria of the Asian Working Group for Sarcopenia was mainly used as the diagnostic criteria. Nineteen studies used muscle strength and muscle mass to diagnose sarcopenia, whereas a full assessment, including physical performance, was performed in five studies. The estimated prevalences of sarcopenia in pre-stroke, within 10 days of stroke, and from 10 days to 1 month after stroke were 15.8%, 29.5%, and 51.6%, respectively. Sarcopenia increased by approximately 15% from pre-stroke to 10 days, and increased by approximately 20% from 10 days to 1 month. Healthcare providers should note that the prevalence of sarcopenia increases during the acute phase in patients with stroke.

## 1. Introduction

Sarcopenia is an age-related skeletal muscle disease that is characterized by a loss of skeletal muscle mass, muscle strength, and physical performance [1]. Sarcopenia is a risk factor for disability, hospitalization, and death [1]. In 2010, the European Working Group on Sarcopenia in Older People (EWGSOP) defined sarcopenia as the loss of skeletal muscle mass, muscle weakness, and physical function [2], leading to a notable increase in sarcopenia research. In 2014, the Asian Working Group for Sarcopenia (AWGS) published cutoff values of low skeletal muscle mass and muscle strength and physical performance for Asian populations [3]. Subsequently, the updated versions of the EWGSOP [4] and AWGS [5] algorithms were published. Research on sarcopenia has increased dramatically since the definitions of sarcopenia were established.

Sarcopenia is caused not only by aging, but also by multiple factors including nutrition, activity, and disease [1]. Nutritional causes of sarcopenia include insufficient protein and energy intake, micronutrient deficiency, malabsorption, and anorexia [1], while causes that relate to low activity levels include bed rest, inactivity, and decreased physical activity. Diseases that cause sarcopenia include bone and joint diseases, as well as chronic and neurological diseases [1]. In addition, hospital admission itself has been proposed as a iatrogenic cause of sarcopenia [1]. It has been reported that approximately 15% of patients develop new onset sarcopenia during hospitalization in acute care hospitals [6]. Patients hospitalized for disease treatment are more likely to develop sarcopenia and to develop severe sarcopenia due to causes such as malnutrition, low activity levels, and disease. Lack of awareness of sarcopenia among health care providers can lead to insufficient energy intake to meet the increased energy requirements caused by inflammation due to disease and surgical invasions; this accelerates the development of iatrogenic sarcopenia.

Globally, stroke is a primary disease, causing death and disability [7], and stroke patients are rapidly aging in recent years [8]. In Japan, which has the most aged population in the world, the mean age of stroke onset has increased during the past 20 years [8]. More than 70% of patients with stroke suffered long-term disability, and increasing age is more likely to result in an unfavorable outcome [9]. Therefore, it is essential for medical professionals to take a geriatric perspective in the treatment of stroke patients.

“Stroke-related sarcopenia” is a concept that indicates the onset or severity of sarcopenia due to stroke [10]. Stroke patients are more likely to develop or become severely sarcopenic due to the atrophy associated with paralysis and disuse, spasticity, inflammation, denervation, reinnervation, impaired feeding, and intestinal absorption [10,11]. Sarcopenia negatively affects clinical outcomes in patients with stroke. The prevalence of sarcopenia in stroke patients in acute hospitals has been reported to be 8.5–33.8% [12,13,14,15,16], and sarcopenia leads to unfavorable outcomes 90 days after stroke [15]. In post-acute rehabilitation hospitals, approximately 50% of patients suffer from sarcopenia on admission [17,18,19,20,21,22,23,24,25,26,27,28,29,30,31,32], and sarcopenia inhibits functional recovery and return to home in patients with stroke [25]. In addition, in several studies, pre-stroke sarcopenia was diagnosed using the SARC-F questionnaire [33,34,35,36], and pre-stroke sarcopenia was a predictor of functional outcome after 3 months [34]. Thus, sarcopenia in pre-stroke and stroke patients has a negative impact on clinical outcomes.

Interventions for stroke-related sarcopenia are needed, but details on the size of the target patient population are unclear. Identifying the trajectory of the prevalence of sarcopenia in stroke patients should clarify the number of potential patients requiring intervention and related healthcare costs. In addition, summarizing relevant criteria can help in standardizing the diagnosis of sarcopenia. This systematic review aimed to identify the trajectories of the prevalence of sarcopenia in the pre- and post-stroke periods and to determine the diagnostic criteria used in stroke patients.

## 2. Materials and Methods

### 2.1. Protocol Registration

This systematic review was performed according to Preferred Reporting Items for Systematic Reviews and Meta-analyses Statement guidelines (PRISMA) [37] and was registered in Figshare [38].

### 2.2. Study Search and Selection

A literature search was conducted by a specialist. To extract the appropriate literature, the search terms were “sarcopenia,” “muscle atrophy,” “stroke,” “hemorrhagic stroke,” “cerebrovascular disorders,” “brain infarction,” and “ischemic stroke.” The detailed search strategy is shown in Appendix A. We searched the literature in six databases: MEDLINE, EMBASE, Web of Science, Cochrane Central Register of Controlled Trials (CENTRAL), CINAHL, and Ichushi-web (in Japanese). The last date of the literature search was 4 July 2022.

Two independent reviewers conducted the primary screening by reviewing the title and abstract of the articles using the Rayyan application (Qatar Computing Research Institute, Doha, Qatar) [39]. Two independent reviewers categorized the articles as included or excluded. Subsequently, the full text of the included articles was reviewed and a third independent reviewer made the final decision if any disagreements occurred, according to the PRISMA guideline.

### 2.3. Eligibility Criteria

We included peer-reviewed full-text articles written in English or Japanese between April 2010 and April 2022. We excluded animal models, qualitative studies, case reports, and conference abstracts. We also excluded systematic reviews, but the adequacy of individual articles was verified. We did not exclude any studies based on the age, sex, or region of the study participants.

### 2.4. Data Extraction

We extracted the study design, number of days from stroke onset to diagnosis of sarcopenia, age, sample size, sex, setting, the number of patients with sarcopenia, and components of sarcopenia diagnosis from the included articles. The number of days from onset to diagnosis of sarcopenia was calculated based on the mean and median values reported by each study.

The primary outcome was a prevalence of sarcopenia. We calculated the estimated prevalences of sarcopenia pre-stroke, within 10 days of onset, and between 10 days and 1 month. We considered this to be the most reasonable and clinically acceptable approach when integrating all included studies, because most of them did not diagnose sarcopenia at a specific time point (e.g., 19.77 ± 7.07 days). We calculated the estimated prevalences of sarcopenia for each period by extracting the sample sizes and the numbers of patients with sarcopenia from all included studies. We drew bubble plots of sarcopenia prevalence with bubble size as the sample size for each study. The secondary outcome was the diagnostic criteria for sarcopenia.

## 3. Results

The initial literature search identified 1887 studies. Subsequently, 260 duplicate studies were excluded, and 1627 studies were included in the primary screening. We then reviewed the full text of 55 studies to determine whether they were eligible for inclusion. Twenty studies were excluded at this stage because they were not original papers, lacked definitions of sarcopenia, and had no indication of disease prevalence. Finally, 35 studies were included (Figure 1).

A summary of the included studies is shown in Appendix A. The study populations ranged in age from approximately 60–80 years, with the exception of one study. Most studies were conducted in Japanese convalescent rehabilitation wards. Twelve studies were conducted in acute care hospitals. 

Of the 35 studies, 32 (91.4%) included Asian patients (Table 1), where 26 (81.3%) included Japanese patients. Two studies (5.7%) were European, and one study (2.9%) was conducted in the USA. Of the 32 studies on Asians, 17 used the AWGS or updated AWGS2019 and 7 studies used the EWGSOP or updated EWGSOP2 to diagnose sarcopenia. Five studies used SARC-F to diagnose sarcopenia. Two European studies used the updated EWGSOP2. The study conducted in the USA used four diagnostic criteria: appendicular lean mass/body mass index, EWGSOP, and International Working Group on Sarcopenia. 

Figure 2 shows the components of the diagnosis of sarcopenia. Nineteen studies used muscle strength and skeletal muscle mass to diagnose sarcopenia. A full assessment of muscle strength, skeletal muscle mass, and physical performance was performed in five studies. Regarding muscle strength measurements for the diagnosis of sarcopenia, 24 of the 30 studies used grip strength, excluding the five studies that used the SARC-F. One study used the Medical Research Council score as a measure of muscle strength. For muscle mass assessment, 24 studies used bioimpedance analysis, four studies used dual-energy X-ray absorptiometry, and one study used calf circumference. Of the five studies that performed a full assessment, one study performed Short Physical Performance Battery (SPBB), three studies used gait speed, and one study used both SPPB and gait speed to define low physical performance.

The combination of muscle mass and strength was the most used parameter for the diagnosis of sarcopenia, followed by the combination of muscle mass, muscle strength, and physical function.

Of a total of 35 studies, 5 studies that used SARC-F and one study that used 4 diagnostic criteria were excluded.

Figure 3 shows the trajectories of the prevalence of sarcopenia in the pre- and post-stroke periods. Most of the included studies were conducted within 40 days of stroke onset; three studies diagnosed pre-stroke sarcopenia using SARC-F. The estimated prevalences of sarcopenia in pre-stroke, within 10 days, and from 10 days to 1 month after stroke were 15.8%, 29.5%, and 51.6%, respectively. The prevalence of sarcopenia increased approximately 15% from pre-stroke to 10 days after onset, and increased by approximately 20% from 10 days to 1 month after onset. 

The bubble size indicates the sample size for the individual studies, and the squares indicate the estimate prevalences of sarcopenia in pre-stroke, within 10 days of onset, and between 10 days and 1 month after stroke, respectively.

## 4. Discussion

We examined the trajectories of the prevalence of sarcopenia in the pre- and post-stroke periods. This systematic review recorded two major findings. First, 91.4% of the studies on stroke-related sarcopenia were conducted in Asian populations, especially in the Japanese population. Accordingly, AWGS or AWGS2019 was mainly used as the diagnostic criteria. Second, the prevalence of sarcopenia in pre-stroke and stroke patients increased over time, it increased by approximately 15% from pre-stroke to 10 days after onset, and increased by approximately 20% from 10 days to 1 month after onset.

East Asia is aging rapidly, especially Japan, which has the highest aging rate worldwide [40]. Indeed, median age of stroke onset has increased over the past 20 years, and the median age of ischemic stroke onset in women has risen to above the age of 80 years since 2016 [8]. The high rate of the aging population may have led to an increasing number studies on stroke-related sarcopenia in Japan, which have mostly been conducted in convalescent rehabilitation wards [17,18,19,20,21,22,23,24,25,26,27,28,29,30,31,32]. Convalescent rehabilitation wards are a unique setting for intensive rehabilitation after the acute phase of stroke [41]. Stroke patients are transferred to convalescent rehabilitation wards approximately 2 weeks after stroke onset when they do not require advanced medical management. Assessing the body composition for sarcopenia is easier during this period because the effects of systemic inflammation, fever, and edema are usually subtle. BIA was used to measure skeletal muscle mass in most of the studies conducted in the convalescent rehabilitation wards. Apart from the non-invasiveness and simplicity of using BIA, the minimal influence of factors inhibiting the measurement of muscle mass may have facilitated the diagnosis of sarcopenia. Combined nutritional management and rehabilitation seems to be an effective intervention for stroke patients with sarcopenia admitted to convalescent rehabilitation wards [42], which may facilitate the assessment of sarcopenia in patients with stroke. In addition, Asian patients with stroke may suffer from sarcopenia due to their small body mass index (BMI) as well as aging. A large discrepancy is observed between the mean BMI of the Asian population, approximately 20 kg/m^2^, and that of people from the Western countries, as demonstrated by a study conducted in the USA included in this review [43] that shows a mean BMI of approximately 30 kg/m^2^. Thus, stroke-related sarcopenia may be a concept focused primarily on the Asian population.

AWGS or AWGS2019 was mainly used to diagnose sarcopenia in Asian patients. However, only five studies diagnosed sarcopenia using a full assessment including muscle strength, skeletal muscle mass, and physical performance, and 19 studies diagnosed sarcopenia using only muscle strength and skeletal muscle mass. Assessment of physical performance in stroke patients is difficult due to impaired consciousness and paralysis, which may be the reason only a few studies performed full assessments. However, it is necessary to discuss the need for a full assessment, including physical performance, since skeletal muscle mass and muscle strength are also directly affected by stroke [44,45]. Alternatively, it is crucial to standardize the diagnostic criteria for sarcopenia in conditions where a full assessment is infeasible.

The prevalence of malnutrition, which is closely related to sarcopenia, has been reported to increase over time from the onset of stroke. In a meta-analysis by Huppertz et al., malnutrition in the hyperacute, early subacute, and chronic phases of stroke increased over time to 19%, 52%, and 72%, respectively [46]. The overall prevalence of undernutrition, not just that in patients with stroke, has been reported to be highest in rehabilitation settings [47,48]. Stroke patients develop muscle atrophy [44] and muscle weakness [45] immediately after the onset of the disease. Muscle weakness and skeletal muscle loss due to paralysis and related disuse [49] reduce physical activity, which leads to the onset or increased severity of sarcopenia. A systematic review showed that more than 78% of time is spent in the sedentary state regardless of the time after the onset of stroke [50]. In addition, consciousness and dysphagia cause decreased nutritional intake, which accelerates the loss of muscle mass and muscle weakness. Dysphagia and sarcopenia affect each other. Patients with dysphagia would have more severe sarcopenia compared with those without [19,51]. Many studies have diagnosed sarcopenia using only muscle strength and muscle mass, which may cause overdiagnosis that has led to the increase in the prevalence of sarcopenia over time. A study that followed the trajectory of sarcopenia indicators after hospitalization for 3 months reported that physical performance recovered early after disease onset, whereas muscle strength and skeletal muscle mass did not recover until the time of hospital admission [52]. The results of this systematic review indicate that sarcopenia is mostly diagnosed by muscle strength and skeletal muscle mass in patients with stroke. The results of our systematic review seem reasonable as the prevalence of sarcopenia increased over time after the onset of the disease.

The strength of this systematic review is that it demonstrates that the prevalence of sarcopenia increases over time in patients with stroke. Identifying the trajectory of the prevalence of sarcopenia can provide a basis for intervention to prevent the onset and worsening of sarcopenia. However, our systematic review has some limitations. First, we could not determine the long-term trajectory of sarcopenia because only a few studies investigated the prevalence of sarcopenia 40 days after stroke onset. Future studies are needed to clarify the long-term trajectory of sarcopenia after the onset of stroke. Second, the timing of sarcopenia diagnosis varied among studies, making it difficult to quantitatively integrate prevalence rates. This seemed to depend on the healthcare systems of the countries in which the studies were conducted.

## 5. Conclusions

This systematic review showed that most studies of sarcopenia in stroke patients have been conducted in Asians, especially Japanese persons. The prevalence of sarcopenia in pre-stroke and stroke patients increased over time. Healthcare providers should be aware that the prevalence of sarcopenia increases during the acute phase of stroke and strive to prevent the onset or worsening of sarcopenia.

## Figures and Tables

**Figure 1 nutrients-15-00113-f001:**
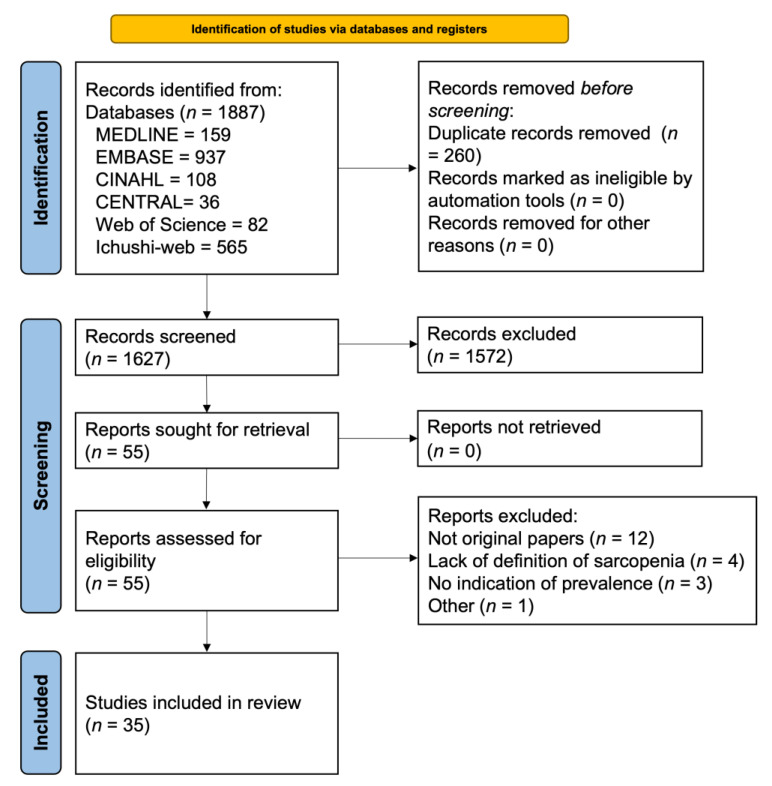
Article selection flowchart according to the Preferred Reporting Items for Systematic Reviews and Meta-analyses Statement.

**Figure 2 nutrients-15-00113-f002:**
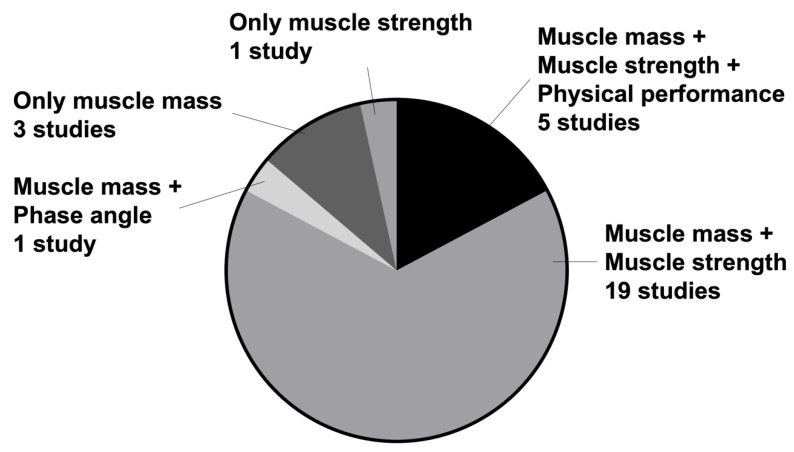
Components of Sarcopenia.

**Figure 3 nutrients-15-00113-f003:**
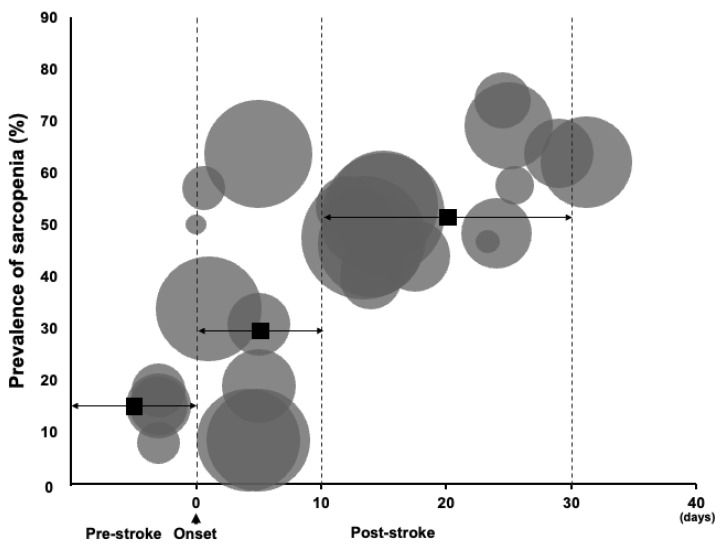
Trajectories of prevalence of sarcopenia in pre- and post-onset period in patients with stroke.

**Table 1 nutrients-15-00113-t001:** Diagnostic criteria or screening tool for sarcopenia according to race.

Diagnostic Criteria (Screening Tool)	Study Region, Number of Studies	
	Asia, *n* = 32	Europe, *n* = 2
AWGS or AWGS2019	17	0
EWGSOP 2	7	2
SARC-F	5	0
FNIH	1	0
Others	2	0

One study that calculated sarcopenia prevalence using multiple diagnostic criteria was excluded from this table. Abbreviations: AWGS, Asian Working Group for Sarcopenia; EWGSOP, European Working Group on Sarcopenia in Older People; FNIH, Foundation for the National Institutes of Health.

## Data Availability

Not applicable.

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
