# Peer review of "Trajectories of the Prevalence of Sarcopenia in the Pre- and Post-Stroke Periods: A Systematic Review"

_nutrients, 2022, doi:10.3390/nu15010113_

Round 1

Reviewer 1 Report

1.      Line 44, the updated version,…

2.      Line 54 to 57, the sentence is supposed to further explain detailed iatrogenic causes that render the patients developing new onset sarcopenia during hospitalization, rather than repeating the general causes again. “Patients hospitalized for disease treatment…. develop severe sarcopenia due to causes such as nutrition, low activity levels…” it makes sense that patients develop sarcopenia during hospitalization for low physical activity, but how come for malnutrition ? in a hospitable?

3.      In data extraction, how the authors defined the period, within 10 days of onset, and between 10 days and 1 month optimal for Trajectory.  

4.      According to line 113-115, “The estimated prevalence was calculated by dividing the sum of the prevalence for each period by the number of studies”. So, in pre-stroke period, the number of studies is equal to 3 (3 bubbles?) and each of the bubble contain different prevalence depending on sample size ?   

Reviewer 2 Report

A agree to the publication in full

Author Response

1